# Structural studies on ribosomes of differentially macrolide-resistant *Staphylococcus aureus* strains

André Rivalta[1], Aliza Fedorenko[1], Alexandre Le Scornet[2], Sophie Thompson[1], Yehuda Halfon[3], Elinor Breiner Goldstein[1], Sude Çavdaroglu[4], Tal Melenitzky[1], Disha-Gajanan Hiregange[1], Ella Zimmerman[1], Anat Bashan[1], Mee-Ngan Frances Yap[2], Ada Yonath[1]

Antimicrobial resistance is a major global health challenge, diminishing the efficacy of many antibiotics, including macrolides. In *Staphylococcus aureus*, an opportunistic pathogen, macrolide resistance is primarily mediated by Erm-family methyltransferases, which mono- or dimethylate A2058 in the 23S ribosomal RNA, reducing drug binding. Although macrolide–ribosome interactions have been characterized in nonpathogenic species, their structural basis in clinically relevant pathogens remains limited. In this study, we investigate the impact of *ermB*-mediated resistance on drug binding by analyzing ribosomes from *S. aureus* strains with varying levels of *ermB* expression and activity. Using cryo-electron microscopy, we determined the high-resolution structures of solithromycin-bound ribosomes, including those with dimethylated A2058. Our structural analysis reveals the specific interactions that enable solithromycin binding despite double methylation and resistance, as corroborated by microbiological and biochemical data, suggesting that further optimization of ketolide–ribosome interactions could enhance macrolide efficacy against resistant *S. aureus* strains.

## Introduction

*Staphylococcus aureus* is a Gram-positive rapidly growing bacterium that normally lives as a human commensal in the skin and the upper respiratory tract (Foster, 2002). In addition, it is an opportunistic bacterial species that may cause severe health problems, with infections involving the skin, the respiratory tract, the urinary system, the bloodstream, and, increasingly, the heart, as *S. aureus* is being a major cause of endocarditis in developed countries (Fowler et al, 2005; Tong et al, 2015). Approximately 20–30% of human adults are persistently colonized by *S. aureus*, and carriage in infancy can approach 70% (Turner et al, 2019; Piewngam & Otto, 2024).

Often acquired in nosocomial or other healthcare settings, its versatility has led over the years to the appearance and spread of several antimicrobial-resistant strains (Stryjewski & Corey, 2014), including MRSA, the methicillin-resistant *S. aureus*. MRSA is a particularly concerning strain that has been listed as a pathogen of high priority by the World Health Organization (WHO) for the development of new antibiotics, along with other ESKAPE pathogens, such as *Enterococcus faecium, S. aureus, Klebsiella pneumoniae, Acinetobacter baumannii, Pseudomonas aeruginosa* (Chang et al, 2022), and has been designated a "serious threat" by the US Centers for Disease Control and Prevention (CDC) (WHO, 2014; CDC, 2019). With the steady emergence of more multi-drug-resistant strains, and with an underwhelming development of new antimicrobial drug classes, a scenario in which bacterial infections prove as dangerous as in the past is getting more credible (Auerbach-Nevo et al, 2016; Matzov et al, 2017b). Resistant bacteria spread through humans, animals, and inanimate objects, allowing them to establish themselves in healthcare settings, communities, and food chains, thus jeopardizing not only vulnerable populations but also the stability of entire health systems (Okeke et al, 2024). In 2019 alone, antimicrobial resistance was responsible for more than 1.2 million deaths worldwide, with *100,000* attributable to just MRSA (Murray et al, 2022; Okeke et al, 2024). Similarly, the largest number of global deaths attributable to antimicrobial resistance in 2021 was ascribed to *S. aureus* infections (Bertagnolio et al, 2024). In *S. aureus*, target protection and removal of antimicrobial compounds are among the most effective resistance mechanisms, as they can counter a wide range of clinically used macrolide, lincosamide, and streptogramin ($MLS_b$) antibiotics and have been identified in both methicillin-sensitive and methicillin-resistant strains (Miklasińska-Majdanik, 2021). Efflux pumps actively expel antimicrobial agents from the bacterial cell, thereby reducing intracellular drug concentrations and playing a critical role in multidrug resistance in Gram-positive bacteria (Schindler & Kaatz, 2016). This resistance can be further complemented by the chemical modification of the target site, a prevalent strategy to hinder the binding of antimicrobial molecules. Within the ribosome, this often

---

[1]Department of Chemical and Structural Biology, Weizmann Institute of Science, Rehovot, Israel    [2]Department of Microbiology-Immunology, Northwestern University Feinberg School of Medicine, Chicago, IL, USA    [3]Astbury Centre for Structural Molecular Biology, University of Leeds, Leeds, UK    [4]Department of Emergency Medicine, Arnavutköy State Hospital, Istanbul, Turkey

Correspondence: ada.yonath@weizmann.ac.il

occurs through various ribosomal RNA (rRNA) methyltransferases, which confer resistance by modifying the specific rRNA residues involved directly or indirectly in the binding of the antimicrobial compounds. These alterations protect while maintaining translational efficiency above lethal thresholds (Jeremia et al, 2023). The posttranscriptional modification of adenosine 2058 (A2058, *Enterococcus coli* numbering used throughout, is A2085 in *S. aureus*) in the 23S rRNA changes the target of several antimicrobial molecules, including MLS$_b$ antibiotics, and is a prominent example of this mechanism. It was first reported in the 1970s (Lai & Weisblum, 1971), and about a decade later, this mechanism was also shown to involve the erythromycin-inducible activity (Gryczan et al, 1984; Dubnau, 1985; Weisblum, 1995b) of a methyltransferase belonging to the Erm family (Shivakumar & Dubnau, 1981; Skinner et al, 1983), which adds specifically one or two methyl groups to A2058, forming either N6-monomethyladenine (m$^6$A2058) or N6,N6-dimethyladenine (m$_2^6$A2058), depending on the bacterial species and the enzyme itself (Weisblum, 1995a). The translation of *erm* genes is regulated by a mechanism involving ribosome stalling induced by a leader peptide, expressed by a leader mRNA sequence located immediately upstream of the coding region. Under normal conditions and in the absence of erythromycin or other MLS$_b$ antibiotics, although the ribosome is translating the *ermBL* leader peptide, the mRNA region containing the ribosome binding site and the *ermB* start codon folds into a stem–loop structure, which impedes the binding to the ribosome and thus the translation of the methyltransferase. Once erythromycin binds the ribosome translating the leader peptide, the ribosome stalls, causing the mRNA to adopt a different secondary structure and exposing the Shine–Dalgarno sequence of the Erm coding region (Ramu et al, 2009; Arenz et al, 2014; Dzyubak & Yap, 2016).

High-resolution structural studies have been instrumental in understanding antibiotic binding in prokaryotic ribosomes. Although previous research demonstrated the macrolide binding and resistance mechanism to ribosomes in the nonpathogenic bacterium *Thermus thermophilus* (Svetlov et al, 2021), it remained unclear how these findings can be related to clinically relevant pathogens. Furthermore, it was suggested that macrolides could bind to methylated ribosomes when paired with a secondary antibiotic such as hygromycin A (Chen et al, 2023). Earlier work on *S. aureus* offered one of the first glimpses of macrolide interactions in pathogens (Eyal et al, 2015), as well as a resistance mechanism based on exit tunnel remodeling (Halfon et al, 2019). Here, we aimed to elucidate, from a structural perspective, the mechanism of *erm*-mediated resistance in a clinically important species such as *S. aureus*, as well as to understand the effects of the methyltransferase activity on various macrolides. In addition, we highlighted how specific moieties used in ketolides may aid in improved binding to the ribosomal site, even in the presence of a modified A2058. Our focus is on solithromycin (SOL), a bactericidal (Svetlov et al, 2017) fourth-generation macrolide (Fernandes et al, 2016; Fernandes et al, 2017), because of its enhanced ribosomal binding properties. Previous chemical probing studies have shown that SOL can interact with dimethylated ribosomes (Llano-Sotelo et al, 2010), making it an ideal candidate for understanding macrolide interactions with methylated ribosomes.

We used *S. aureus* strains harboring various engineered mutated *ermBL-ermB* cassettes, which were shown to affect both methyltransferase activity and translation. These mutations target the *ermBL* leader peptide, as well as the *ermB* coding sequence itself (Table 1) (Shields et al, 2024). These engineered strains exhibited varying levels of ErmB expression and/or activity, with some displaying overexpression and others combining overexpression with hyperactivity (e.g., MNY196). Minimum inhibitory concentration tests performed on these strains showed a positive correlation between higher expression and/or higher activity of ErmB and the concentration of the antibiotic required to prevent bacterial growth (Table 1). Thus, our structural study provides a unique model to elucidate the role of overexpression and hyperactivity of ErmB in a clinically significant ESKAPE pathogen, such as *S. aureus*, while also confirming the binding mechanism of MLS$_b$ antibiotics to the universally conserved nucleotide A2058 (*E. coli* numbering used throughout unless otherwise indicated).

## Results

### Half-maximal inhibitory concentration (IC50) estimation

Aiming to understand whether the increased resistance was correlated to reduced binding of the drugs to its ribosomal target, or to other mechanisms present in the living bacterial cell (e.g., reduced uptake and/or increased efflux), we performed luciferase activity assays in an in vitro coupled transcription/translation system using ribosomes isolated from various ErmB-containing strains (Table 1). These assays helped us to assess the IC50 estimates of the macrolides being tested, that is, the concentration required to inhibit the ribosomal activity by 50%.

Our assays showed that strains exhibiting higher expression or activity levels of ErmB displayed increased resistance across all tested macrolides.

Ribosomes from the MNY196 strain, with the highest expression levels and ErmB catalytic activity, were most resistant to all the macrolides tested. In contrast, strains with the low or basal expression of ErmB exhibited significant sensitivity to the same antibiotics (Table 2, Fig S1).

### Cryo-EM structures of 70S ribosomes from three *S. aureus* (SA) strains exhibiting differential SOL resistance

To decipher the binding mechanism of SOL in the context of *S. aureus* ribosome, we purified the ribosomes from cells of the various strains grown to the mid-log phase. The sucrose gradient profiles revealed two peaks, and we selected the first one, corresponding to the 70S ribosome population (similar to Shields et al [2024]), for cryo-EM single-particle analysis (Table 3, Fig S2).

The unmodified, macrolide-sensitive SA ribosome purified from strain KES34 and complexed with solithromycin (SA_KES34_SOL) before vitrification was imaged on a Titan Krios using a Gatan K3 camera with an energy filter. After data processing in RELION, which included CTF refinement (accounting for high-order aberrations, anisotropic magnification estimation, and defocus fitting) and Bayesian per-particle polishing, we obtained a 2.15 Å cryo-EM

**Table 1.   Various strains used in our study are derived from *S. aureus* USA300 JE2, which natively lacks erm genes.**

| Strain | Genotype | Description | MIC (µg/ml) | | |
|---|---|---|---|---|---|
| | | | ERY | AZI | SOL |
| KES29 | *ermBL*(WT), *ermB* (WT) | Basal expression of *ermB* without ERY | 12 | 48 | 2 |
| KES34 | *ermBL*(WT), *ermB* (Y103A) | Catalytically inactive ErmB | 0.095 | 0.125 | 0.125 |
| MNY196 | *ermBL*(R7stop), *ermB* (I75T/N100S) | Overexpression and hyperactive ErmB | >256 | >256 | >32 |

The differential expression and activity of ErmB of these strains correlate to their minimum inhibitory concentration (MIC) values against macrolides, as reported in Shields et al (2024).

**Table 2.   IC50 estimates (in µM) of erythromycin (ERY), azithromycin (AZI), and solithromycin (SOL) against ribosomes from strains KES34, KES29, and MNY196 ± standard error of n > 2 independent replicates.**

| Strain | ERY | AZI | SOL |
|---|---|---|---|
| KES29 | 0.4 ± 0.08 | 0.4 ± 0.06 | 0.31 ± 0.03 |
| KES34 | 0.43 ± 0.04 | 0.33 ± 0.04 | 0.47 ± 0.02 |
| MNY196 | >250 | >250 | >180 |

reconstruction of the 70S ribosome (resolution values throughout calculated on the postprocessed, masked maps using the gold-standard FSC 0.143 threshold, Fig S3). Although tRNA molecules and mRNA nucleotides were not intentionally assembled into the complex, their high-quality density allowed us to model these molecules, thereby underscoring the functional integrity of the purified ribosomes (Fig S4). To improve resolution and to aid in modeling, we performed a multibody refinement focused on the large ribosomal subunit and the small subunit (head and body), which yielded a 2.07 Å map for the 50S subunit, and 2.38 Å maps for both the small subunit head and the body. Remarkably, the cryo-EM map revealed additional densities corresponding to P-site and E-site tRNAs, along with fragments of an A-site tRNA and mRNA. Within the nascent peptide exit tunnel (NPET), solithromycin is clearly resolved, binding through key interactions with conserved rRNA residues across bacterial species (Fig 1A and B). Similar to other macrolides, the desosamine group of the antibiotic establishes water-mediated H-bonds with the N6 of A2058 and with the phosphate group of G2505, and a direct H-bond with the N2 of A2058. In addition, the alkyl–aryl side chain of SOL appears to have stacking interactions with the rRNA base pair U2609:A752. The high resolution achieved, in addition, allowed us to model several rRNA modifications (Table S1), including $m^7$G2574 (G2601 in *S. aureus* numbering) in helix H90 of the 23S rRNA, which was previously reported on *S. aureus* (Bahena-Ceron et al, 2024), and to identify specific Zn binding isoforms of ribosomal proteins (including bL32, eL33 type 1, eL36).

Because it was established that SOL binds dimethylated ribosomes in *S. aureus* (Llano-Sotelo et al, 2010), and a prior study proposed that the weak binding observed for another ketolide, TEL, to dimethylated ribosomes was due to the presence of a subpopulation of unmethylated ribosomes in the sample (Chen et al, 2023), we aimed to investigate whether SOL can indeed bind directly to fully dimethylated ribosomes.

Hence, we determined the cryo-EM structure of the ribosome–SOL complex from MNY196, which features both hyperproduced and hyperactive ErmB. The data were collected and processed under conditions similar to those of SA_KES34_SOL, yielding a 2.15 Å cryo-EM reconstruction of the 70S ribosome. Multibody refinement focused on the large subunit revealed at 2.08 Å resolution (Fig S5) that the antibiotic can be accommodated into the ribosome at high concentrations and bind its pocket, despite the double methylation of A2058 (Fig 1B). Importantly, the density corresponding to the two methyl groups of A2058 is clearly resolved, confirming their presence and suggesting that they contribute to the subtle displacement of the coordinating water molecule. The density of the antibiotic is also clearly defined, indicating stable binding within the ribosome. Note that similar to SA_KES34_SOL, some residues of the bound tRNA molecules could be resolved.

Comparative analysis of the NPET using the PoseEdit 2D interaction maps (Fig 2A and B) shows that SOL establishes a nearly identical contact network in the unmethylated (SA_KES34_SOL) and dimethylated (SA_MNY196_SOL) ribosomes: the main difference is the loss of a water-mediated hydrogen bond to A2058 when the nucleotide is dimethylated, owing to a 1.1 Å shift of the water molecule (Fig 2C). Despite this change, critical stabilizing interactions are retained, such as those in the proximity of the fluorine atom to C2611 (Fig 2D) and the aromatic–aromatic (π–π) interaction with the A752:U2609 base pair (Fig 2E). These interactions appear sufficient to stabilize SOL within the binding site, even in the presence of a dimethylated A2058.

## Discussion

### Resistance to macrolides

Our results from the cell-free luciferase activity assays indicate that the increased macrolide resistance observed in various SA strains is indeed inversely linked to reduced ribosomal inhibition by macrolides, as shown by the higher IC50 values in strains with the higher expression and/or activity levels of ErmB. MNY196, in which ErmB exhibited the highest activity, resulting in up to 47–51% of methylated ribosomes (Shields et al, 2024), showed the most notable reduction in drug-induced ribosomal inhibition, supporting the role of ribosomal modifications in resistance. These results also strongly correlate to the minimum inhibitory concentration values (Shields et al, 2024), suggesting that reduced macrolide binding to the ribosome because of the ErmB methylation is a key driver of resistance.

Notably, the ErmB mutations in MNY196 (I75T and N100S), previously reported to confer resistance to TEL (Al-Lahham et al, 2006), could also be associated with resistance to SOL (Shields et al, 2024).

**Table 3.  Details of cryo-EM data collection and processing.**

| | SA_KES34_SOL | SA_MNY196_SOL |
|---|---|---|
| Microscope | Titan Krios | Titan Krios |
| Camera | Gatan K3 | Gatan K3 |
| Energy filter (slit eV) | GIF energy filter (15) | GIF energy filter (15) |
| Voltage (kV) | 300 | 300 |
| Magnification | 105K | 105K |
| Pixel size (Å/px) | 0.793 | 0.793 |
| Calibrated pixel size (Å/px) | 0.824 | 0.824 |
| Defocus range (μm) | (−1.8) − (−0.5) | (−1.8) − (−0.5) |
| Dose rate (e−/px/s) | 17.9 | 18 |
| Electron dose per frame (e−/Å) | 1.00 | 1.00 |
| Micrographs collected | 4,053 | 5,120 |
| *Refinement* | | |
| No. of particles autopicked | 564,732 | 397,842 |
| No. of particles in final 3D reconstruction | 408,831 | 263,087 |
| Resolution-unmasked consensus map (Å) | 2.34 | 2.41 |
| Resolution solvent–masked consensus map after sharpening (Å) | 2.145 | 2.158 |
| Resolution LSU map after postprocessing (Å) | 2.071 | 2.08 |
| Resolution SSU map after postprocessing (Å) | NA | 2.37 |
| Resolution SSU body after postprocessing (Å) | 2.38 | NA |
| Resolution SSU head map after postprocessing (Å) | 2.38 | NA |
| *Modeling and validation* | Composite 70S | LSU |
| Model-to-map resolution, unmasked (FSC = 0.5) | 2.2 | 2.2 |
| Nonhydrogen atoms | 130,292 | 82,292 |
| Protein residues | 5,198 | 3,027 |
| Nucleotide residues | 4,249 | 2,765 |
| Bonds (RMSD), length (Å) (# > 4σ) | 0.004 (0) | 0.005 (0) |
| Bonds (RMSD), angles (°) (# > 4σ) | 0.643 (26) | 0.741 (5) |
| MolProbity score | 1.85 | 1.65 |
| Clash score | 4.94 | 4.07 |
| *Protein geometry* | | |
| Ramachandran favored (%) | 96.61 | 96.79 |
| Ramachandran allowed (%) | 3.39 | 3.21 |
| Ramachandran outliers (%) | 0.00 | 0.00 |
| Rotamer outliers (%) | 3.07 | 2.23 |
| *RNA geometry* | | |
| Correct sugar puckers (%) | 99.81 | 99.57 |
| Correct bonds (%) | 100 | 100 |
| Correct angles (%) | 100 | 100 |
| Correct backbone conformations (%) | 87.88 | 88.09 |

Map-to-map resolution calculated at FSC = 0.143 (gold standard). FSC = Fourier shell correlation. Postprocessing here refers to solvent-masked map sharpening as implemented in RELION.

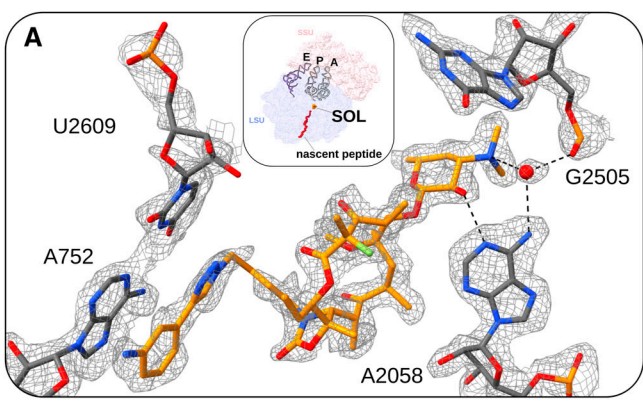

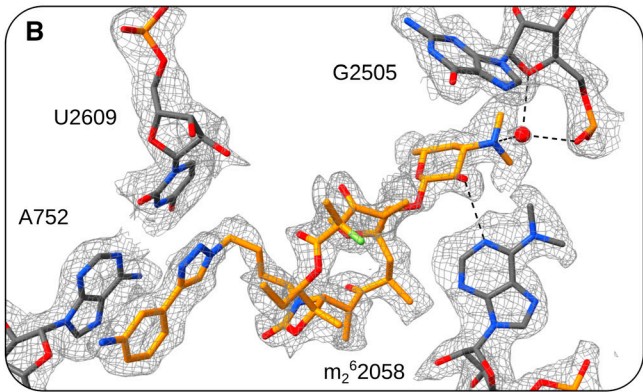

**Figure 1. Binding pocket of solithromycin (SOL) in two SA strains.**
**(A)** Inset: full view of the ribosome (with the small subunit in light pink and the large subunit in deep blue) showing the location of SOL (dark orange) bound in the NPET within the large subunit, with the tRNA molecules displayed as cartoons and a model of the nascent peptide chain shown in red. **(A)** Close-up view of the antibiotic binding site, the unmethylated A2058 residue in SA_KES34_SOL. **(B)** Dimethylated A2058 residue in SA_MNY196_SOL. SOL is depicted in dark orange, the coordinating water molecule as a red sphere, and hydrogen bonds as black dashes. The cryo-EM maps in (A, B) are shown as a gray mesh, with contour levels of $2.5\sigma$ and $1.9\sigma$, respectively. Upon dimethylation of A2058, hydrogen bonds between the coordinating water molecule and the N6 position of the base are lost.

Despite these functional disparities, our cryo-EM maps reveal only minor local adjustments within the tunnel, indicating that resistance arises mainly from weakened contacts rather than whole structural rearrangements.

## Solithromycin binds stably even in the presence of a double-methylated A2058

Solithromycin, the first fluoroketolide, is considered a fourth-generation macrolide owing to its improved activity against telithromycin-resistant strains (Fernandes et al, 2016). Despite its clinical use and the emergence of resistant clinical isolates (Fernandes et al, 2016; Yao et al, 2019), no 3D structure of this antibiotic bound to a clinically important pathogen has been reported. Our study shows that the interactions of SOL in the *S. aureus* ribosome align closely with the previously published structures of the *E. coli* ribosomes (Llano-Sotelo et al, 2010), validating the common nature of the SOL binding mode in bacteria (Fig 3).

Our study demonstrates that SOL can bind to the ribosome both in unmodified and in double-methylated states. Although methylation at A2058 causes the displacement of the coordinating water molecule between N6 of the nucleotide and the desosamine moiety, the binding of this ketolide to the ribosome seems unaffected, at least pertaining to the cryo-EM sample preparation conditions. When comparing the double-methylated ribosome to the unmodified one, we observe that the SOL molecule does not appear to shift, but the coordinating water molecule does, along with a slight displacement of the A2058 residue (Fig 2). In SA_MNY196_SOL, the strong SOL density in the cryo-EM map suggests that SOL binds directly to the dimethylated ribosome rather than selectively interacting with a subpopulation of unmethylated or partially methylated ribosomes as suggested by Chen et al (2023) relating to TEL binding.

The partial retention of the water bridge contrasts sharply with findings in *T. thermophilus* ribosomes, where double methylation was shown to completely disrupt the water-mediated interaction and prevents ERY or TEL from binding unless hygromycin A, a PTC inhibitor, is present (Svetlov et al, 2021; Chen et al, 2023). This discrepancy highlights the importance of studying clinically relevant pathogens, as structural and functional observations in model organisms like *T. thermophilus* may not fully recapitulate the complexities of resistance mechanisms in human pathogens.

### Structural comparison with erythromycin

Both ERY and SOL, as other macrolides, bind in the ribosomal NPET with aligned macrolactone rings (Fig 4C), but SOL differs from ERY in three major aspects (Fig 4A and B). First, the cladinose sugar is replaced by a keto group (as in other ketolides), reducing the molecule's overall bulk near the PTC. Second, an additional fluorine atom enables a hydrophobic contact with C2611. Third, an alkyl–aryl side chain extends from the macrolactone scaffold, creating additional π-stacking and van der Waals contacts unavailable to erythromycin (Fig 4C). Together, these features may explain why SOL remains more stably anchored without the addition of a secondary antibiotic, in contrast to previous research (Chen et al, 2023). As in the case with TEL, the alkyl–aryl side chain of SOL establishes an essential interaction with the U2609:A752 base pair, which was shown to be important for ribosome functionality and for establishing the unique interactions with the extended side chains of ketolides, as well as one of the reasons for the cidality of such antibiotics. Binding kinetics studies in *Streptococcus pneumoniae* suggested that double methylation of A2058 accelerates the dissociation of SOL from the ribosomes, reducing its bactericidal activity, even though cells experienced translation inhibition (Svetlov et al, 2017; Svetlov et al, 2020). Nonetheless, once SOL is bound, it dissociates markedly more slowly than macrolides such as ERY, a property that can be attributed to its additional anchoring points in the binding pocket. The same slow dissociation rate that underlies SOL's bactericidal action in *S. pneumoniae* is likely a key factor in *S. aureus* as well, supporting the notion that SOL's slow dissociation enables stable binding even under resistance-inducing modifications.

Although SOL requires further safety studies, our findings underline its value as a structural prototype for designing next-generation antibiotics. Features such as its ability to engage resistant ribosomes provide critical insights into how drug design might circumvent common resistance mechanisms, including ribosomal methylation.

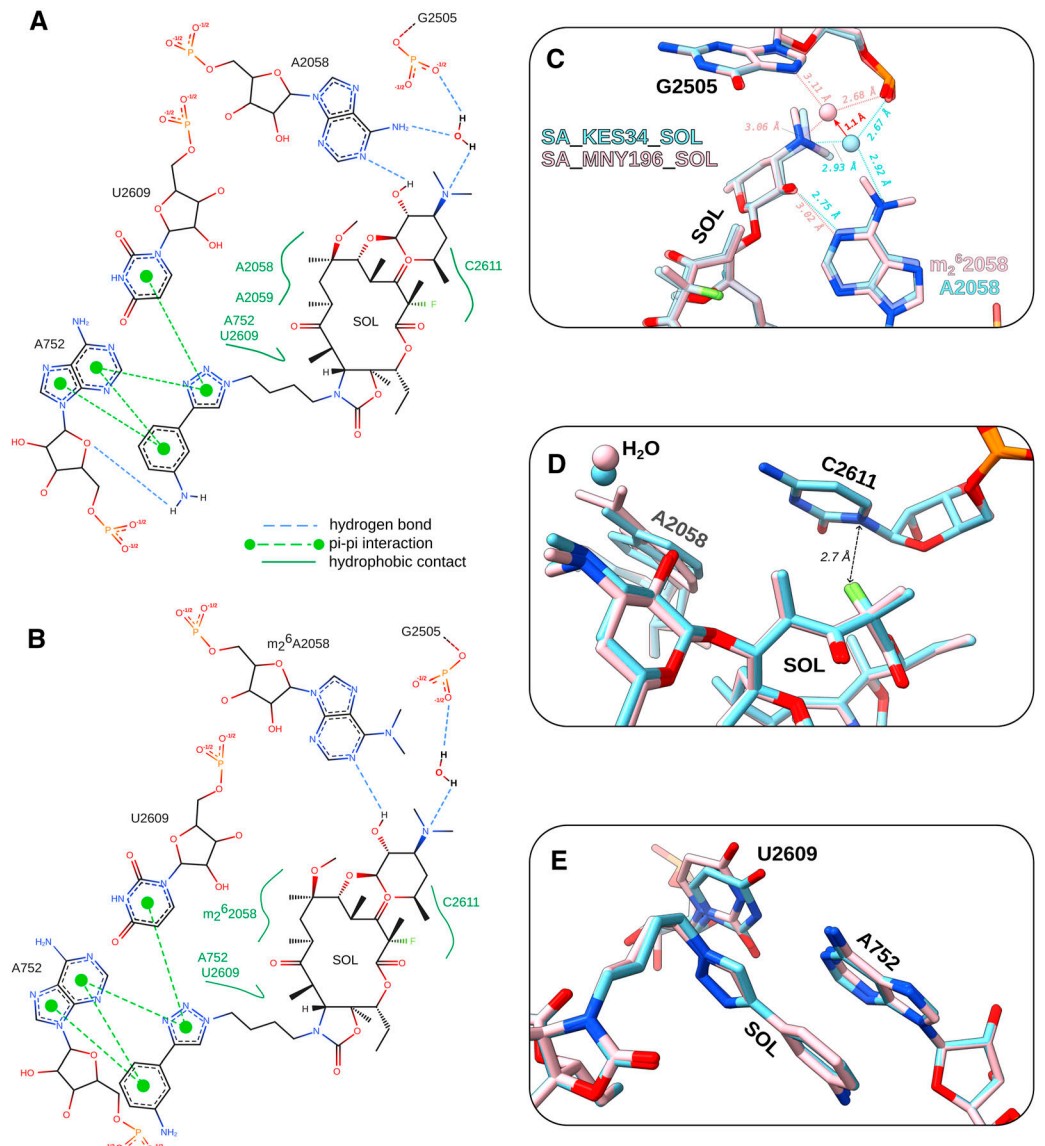

**Figure 2.   Structural comparison of SOL-bound KES34 and MNY196 ribosomes.**
**(A, B)** PoseEdit diagrams illustrating the interactions of SOL with its surrounding environment; H-bonds are depicted as blue dashed lines, hydrophobic contacts as dark green curved lines, and pi-stacking interactions as light green dashed lines in SA_KES34_SOL (A) and SA_MNY196_SOL (B). The coordinating water molecule and its H-bonds were added manually for reference. **(C, D, E)** Superposition of SA_KES34_SOL (sky blue) and SA_MNY196_SOL (pink), with detailed interactions highlighted. Hydrogen bonds calculated using ChimeraX and Coot are depicted as dashed lines and distances as dotted, double-headed arrows. The root-mean-square deviation of the superposition between SA_MNY196_SOL and SA_KES34_SOL is < 0.7 Å. **(C)** Water-mediated interaction of the desosamine group of SOL with the A2058 residue (unmethylated in SA_KES34_SOL) and the phosphate group of G2505. The water molecule is depicted as a sphere, pink for SA_MNY196_SOL and sky blue for SA_KES34_SOL. A red arrow indicates the 1.1 Å shift in water position between the unmethylated and methylated state. The direct hydrogen bond between the desosamine group of SOL and the N1 of A2058 elongates from 2.75 Å in the unmethylated state to > 3 Å in the dimethylated state, indicating a weaker interaction. **(D)** Fluorine atom of the SOL sits at a short distance from C2611, facilitating drug binding. **(E)** Aromatic–aromatic (π-stacking) interaction between the alkyl–aryl side chain of SOL and the rRNA base pair U2609-A752.

In conclusion, our work enhances the understanding of macrolide interactions with MLS-resistant ribosomes. By demonstrating solithromycin's capacity to bind in resistant strains, these findings provide a foundation for further exploration of both resistance dynamics and the development of novel antibiotics that exploit structural flexibility to achieve efficacy in the face of resistance.

# Materials and Methods

## *S. aureus* ribosome purification

### Bacterial strains and chemicals
Strains used in this experiment were described in Shields et al (2024). Solithromycin was custom-synthesized by LGM Pharma.

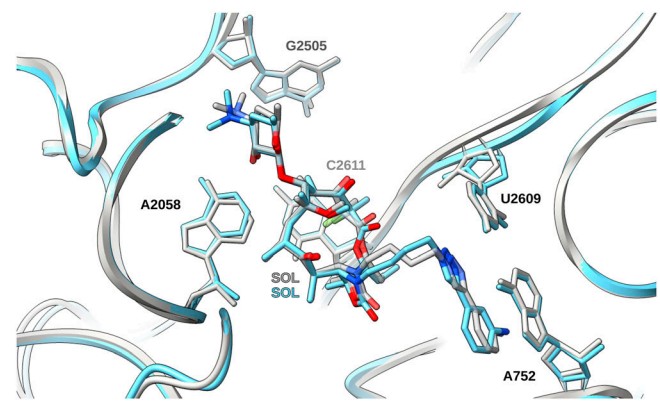

**Figure 3. Superposition of the cryo-EM structure of SA_KES34_SOL (sky blue) and of *E. coli* in complex with SOL (gray, PDB 4WWW), with A2058 (RMSD of the superposed structures < 0.85 Å).**
Solithromycin appears to bind in *S. aureus* in a similar manner to *E. coli*.

Erythromycin and azithromycin were purchased from Sigma-Aldrich.

### Bacterial growth, harvest, and cell lysis

The stabs of each strain were revived and grown in tryptic soy broth (TSB) media with 0.15 mM $CdCl_2$, and glycerol stocks were prepared. Each strain was grown overnight (O/N) at 37°C on a TSB plate containing 0.15 mM $CdCl_2$. A single colony was picked and used to inoculate a 2 ml starter media (supplemented with 0.15 mM $CdCl_2$) and grown for 8 h, and the turbid media were transferred into a larger vessel containing 100 ml of TSB, 0.15 mM $CdCl_2$. The bacteria were grown O/N, and 10 ml was transferred into a 2-liter baffled Erlenmeyer flask containing 1 liter of TSB and grown until an $OD_{600}$ ~1.5. The bacterial culture was transferred into 1-liter centrifuge tubes and centrifuged for 15 min/6,000 rpm/4°C using a Fiberlite F10-4x1000 LEX rotor (Thermo Fisher Scientific). The pellet was resuspended in S30 Buffer A (10 mM Tris acetate, pH 8, 14 mM Mg acetate, 1 M KCl, 1 mM DTT) and centrifuged again under the same conditions. The supernatant was discarded, and the pellet was resuspended in Buffer A (10 mM Tris acetate, pH 8, 14 mM Mg acetate, 50 mM KCl, 1 mM DTT) and centrifuged under the same conditions. The supernatant was discarded, and the pellet was resuspended in a small amount of Buffer A and centrifuged in a tabletop centrifuge. Lysis was performed as previously described (Eyal et al, 2015).

### Ribosome purification

The following buffers were used for the ribosome purification, filtered using 0.2-$\mu$m filters, and kept on ice or at 4°C; when present, beta-mercaptoethanol (BME) was added just before use: sucrose cushion buffer (1.1 M RNase-free sucrose, 60 mM KOAc, 14 mM Mg [OAc]$_2$, 10 mM Tris, pH 8, 6 mM BME); 0% gradient buffer (10 mM Hepes, pH 8, 15 mM $MgCl_2$, 100 mM $NH_4Cl$, 50 mM KCl, 6 mM BME); 10% gradient buffer (10% wt/vol RNase-free sucrose, 10 mM Hepes, pH 8, 15 mM $MgCl_2$, 100 mM $NH_4Cl$, 50 mM KCl, 6 mM BME); 40% gradient buffer (40% wt/vol RNase-free sucrose, 10 mM Hepes, pH 8, 15 mM $MgCl_2$, 100 mM $NH_4Cl$, 50 mM KCl, 6 mM BME); ribosome storage buffer (10 mM Hepes, pH 7.6, 10 mM $MgCl_2$, 60 mM $NH_4Cl$, 15 mM KCl).

The total volume of the cell lysate was divided into two 26.3-ml polycarbonate ultracentrifuge bottles (Beckman Coulter), and 5 ml of cold sucrose cushion buffer was gently injected at the bottom of each tube. Samples were ultracentrifuged for >17 h in a Type 70 Ti rotor (Beckman Coulter) at 55,000 rpm, 4°C. The supernatant was discarded, and the pellet was briefly washed and eventually resuspended with 0% gradient buffer; ribosome concentration was determined in $A_{260}$ units with a NanoPhotometer (Implen).

For the sucrose gradient preparation, six 38.5-ml Open-Top Thinwall Ultra-Clear Tubes (Beckman Coulter) were loaded with 10% and 40% gradient buffer, and the gradients were made using BioComp Gradient Master. About 150 $\mu$L of dissolved ribosomal particles at a concentration of about 80 $A_{260}$ units from the previous step was gently added on top of each tube, and samples were then ultracentrifuged in a SW-28 swinging bucket rotor (Beckman Coulter) at 19,000 rpm for 14 h at 4°C.

Ribosomal fractions were collected using BioComp Piston Gradient Fractionator coupled with a Bio-Rad Econo UV monitor (254-nm filter) to detect the ribosomal peaks. Two peaks were identified and separately collected during the process for each of the six tubes; similar fractions were pooled together and ultracentrifuged for 19 h in a Type 70 Ti rotor at 55,000 rpm, 4°C. Supernatants were discarded and the pellets resuspended using ribosome storage buffer, to reach a final volume of 1 ml per sample, and ultracentrifuged again in a desktop ultracentrifuge using a TLA-100 fixed-angle rotor (Beckman Coulter) at 70,000 rpm, 2.5 h, 4°C. The supernatant was discarded, and ~200 $\mu$l of ribosome storage buffer was added to allow resuspension O/N at 4°C. A260 values were measured on the NanoPhotometer, and the samples were flash-frozen with liquid $N_2$ to a concentration lower than 1,000 $A_{260}$ and stored at −80°C.

### IC50 measurement

Ribosomal activity was assessed with an in vitro coupled transcription/translation system measuring the expression of the luciferase gene. The assay was performed on a white poly-carbonate 96-well plate (Nunc), with a final volume of 30 $\mu$l in each well. The reaction mix included 0.22 vol/vol *E.coli* cell extract (w/o the ribosomal components), 300 nM of 70S, 3.5 ng/$\mu$l luciferase plasmid, 900 $\mu$M of each amino acid (Sigma-Aldrich), 230 $\mu$g/ml creatine kinase from rabbit muscle (Roche), 150 $\mu$g/ml *E. coli* tRNA mixture (Sigma-Aldrich), 50 mM Hepes-KOH (pH 7.5), 9% PEG 8000, 180 mM KGlu, 1.6 mM DTT, 1.1 mM ATP, 0.8 mM GTP, 0.8 mM CTP and 0.8 mM UTP, 0.6 mM cAMP, 70 mM creatine phosphate, 30 $\mu$g/ml folinic acid, 25 mM $NH_4OAc$, and 16 mM $Mg(OAc)_2$. The 30 $\mu$l reaction mixtures in each well were incubated for 60 min at 37°C in the presence of different concentrations (serial dilutions) of the various antibiotics. To perform the translation quantification, 50 $\mu$l of Luciferase Assay System (Promega) was added into each well, followed by immediate photoluminescence reading using a SpectraFluor Plus plate reader (Tecan) or a CLARIOstar Plus plate reader (BMG LABTECH).

Data were analyzed and visualized using R (R Core Team, 2024), with the *tidyverse* (Wickham et al, 2019) and the *drc* (Ritz, 2010; Ritz et al, 2016, 2019) packages, following published guidelines and recommendations (Sebaugh, 2011; Weimer et al, 2012).

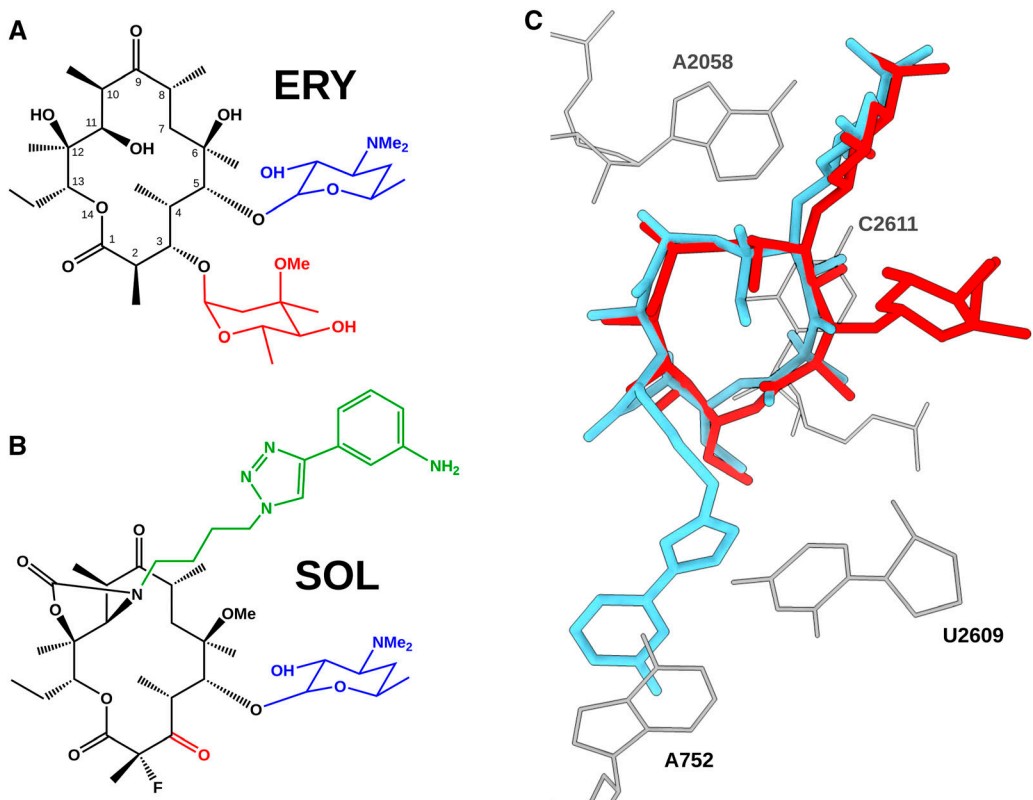

**Figure 4. Structural comparison of solithromycin (SOL) and erythromycin (ERY); macrolactone ring atoms numbered according to the ERY scaffold.**
**(A, B)** 2D chemical structures of the two molecules. The desosamine is highlighted in blue, the C3 substituent in red (cladinose in ERY, keto group in SOL), and the alkyl–aryl side chain unique to SOL in green. SOL also possesses an extra fluorine atom on C2. **(C)** Superposition of SOL in *S. aureus* (SA_KES34_SOL, sky blue) and ERY (from PDB 6S0Z, gray); residues A2058, C2611, A752, and U2609 are shown as sticks. The macrolactone rings largely coincide, but SOL's alkyl–aryl side chain extends away from the core scaffold, and its keto group replaces ERY's bulky cladinose.

## Ribosome structure determination

### Cryo-EM sample preparation

For each strain, the first peak collected from the 70S ribosome purification sucrose gradient was selected for cryo-EM. For the ribosome–SOL complex, 23 µl of 7 $A_{260}$ units of 70S ribosomes (corresponding to about 0.16 µM) were incubated with 2 µl of 1.25 mM SOL (dissolved in 12.5% EtOH), for 30 min at room temperature before vitrification. About 3.5 µl of each sample were flash-frozen using Vitrobot Mark IV (Thermo Fisher Scientific) on glow-discharged Quantifoil R2/2 grids with continuous thin carbon support (2 nm).

### Cryo-EM data collection and processing

For SA_KES34_SOL and SA_MNY196_SOL, cryo-EM data were collected at liquid nitrogen temperature on a Titan Krios electron microscope (Thermo Fisher Scientific) operating at 300 kV, using EPU (Thermo Fisher Scientific). Multiframe micrographs were recorded on a Gatan K3 direct electron detector with an energy filter slit set to 15 eV at a nominal magnification of 105 K with a pixel size of 0.793 Å/pixel in fringe-free mode (later calibrated to 0.824 Å/px) and a dose rate of 1.00 electrons/Å²/frame; defocus values ranged from −0.5 to −1.8 µm. RELION 3.1 and RELION 5 (Scheres, 2012; Zivanov et al, 2020; Kimanius et al, 2021; Nakane & Scheres, 2021) were used

for every data processing step. Micrographs were gain-corrected, aligned, and dose-weighted using RELION's own implementation of MotionCor2 (Zheng et al, 2017); CTF parameters were estimated with CTFFIND4 (Rohou & Grigorieff, 2015). Particles were picked by using a semi-automated approach (Laplacian-of-Gaussian–based auto-picking of a small subset of micrographs, followed by 2D classification of the extracted particles, which were used as references for autopicking in the whole dataset), and then subjected to unsupervised 3D classification using a cryo-EM map of the *S. aureus* ribosome as initial reference (EMD-10079). For SA_MNY196, a 2D classification step was used after particle picking and before 3D classification to remove obvious junk. All classes appearing to contain well-formed 70S ribosomal particles were selected and used for 3D auto-refinement. The refined particles were then subjected to CTF refinement (accounting for higher order aberrations, anisotropic, per-particle defocus), Bayesian per-particle polishing, and again 3D refinement, yielding a 70S reconstruction (consensus map). A multibody refinement step with separate masks manually generated from the consensus map using UCSF ChimeraX for the 50S subunit, the 30S head, and 30S body (or only 30S subunit for SA_MNY196_SOL) was used to improve the details of the PTC and the NPET. Average map resolutions were determined using the gold-standard FSC = 0.143 criterion of solvent-masked, post-processed maps as implemented in RELION. Local resolution was

calculated using relion_postprocess within RELION. UCSF Chimera (Pettersen et al, 2004) and UCSF ChimeraX (Pettersen et al, 2021) were used throughout for map and model visualization, map fitting, and aiding in pixel size calibration.

### Model building

Model building of rRNA and RPs was performed combining template-guided and de novo model building in Coot (Emsley et al, 2010). The coordinates of the *S. aureus* ribosome (PDB 6HMA and 5NGM [Matzov et al, 2017a]) were used as an initial template for model building and were fitted onto EM maps using UCSF Chimera or UCSF ChimeraX, whereas PDB coordinates from 7NHM (Crowe-McAuliffe et al, 2021) and 7K00 (Watson et al, 2020) were used to identify and model A-site, P-site, and mRNA molecules. Model refinement was performed using an iterative approach including real-space refinement and geometry regularization in Coot, followed by real-space refinement and validation using the PHENIX suite (Word et al, 1999; Afonine et al, 2018a, 2018b; Williams et al, 2018; Liebschner et al, 2019) and Servalcat (Yamashita et al, 2021), which aided in identifying unmodeled regions and misplaced atoms. Water molecules, and magnesium and potassium ions were added manually using Find Waters and Check Waters in Coot and considering coordination shells. For SA_KES34_SOL, a composite map, to better model the tRNA molecules, was generated using the PHENIX Combine Composite Maps tool using the postprocessed maps as inputs. The final model was validated using MolProbity (Williams et al, 2018). UCSF Chimera and UCSF ChimeraX were used throughout for map and model visualization.

### Figures

Figures showcasing maps and models were generated using UCSF ChimeraX. The 2D intermolecular interactions were generated with PoseEdit (Diedrich et al, 2023) and edited manually to account for the water molecules using Inkscape. Inkscape was used throughout to assemble the images.

## Data Availability

The cryo-EM density maps have been deposited in the Electron Microscopy Data Bank (EMDB) under the following accession numbers: for SA_KES34_SOL, EMD-52642 (consensus map), EMD-52647 (LSU-focused refinement), EMD-52648 (SSU body–focused refinement), EMD-52649 (SSU head–focused refinement), and EMD-53066 for the combined focused map; for SA_MNY196, EMD-53067 (LSU-focused refinement map). Atomic coordinates have been deposited in the Protein Data Bank (PDB) under accession codes 9QEG and 9QEH for SA_KES34_SOL and SA_MNY196_SOL, respectively.

## Supplementary Information

## Acknowledgements

We thank the members of the A Yonath (in particular, Moshe Peretz and Shoshana Tel-Or) and M-NF Yap groups, as well as Nadav Elad from the Weizmann Electron Microscopy Unit and Yoav Barak from the BioNano Unit, for their interest and experimental support. This work was supported by the Kimmelman Center for Macromolecular Assemblies. A Yonath holds the Martin S and Helen Kimmel Professorial Chair at the Weizmann Institute of Science. The studies performed in the USA were supported by the National Institutes of Health R01AI150986 (to M-NF Yap) and in part by the Department of Defense W81XWH-18-1-0122 (to M-NF Yap).

### Author Contributions

A Rivalta: data curation, validation, investigation, visualization, methodology, and writing—original draft.
A Fedorenko: data curation and investigation.
A Le Scornet: data curation and investigation.
S Thompson: data curation.
Y Halfon: data curation and investigation.
E Breiner Goldstein: data curation and investigation.
S Çavdaroglu: data curation.
T Melenitzky: data curation.
D-G Hiregange: data curation and investigation.
E Zimmerman: data curation and investigation.
A Bashan: conceptualization, supervision, validation, investigation, project administration, and writing—original draft.
M-NF Yap: conceptualization, supervision, funding acquisition, and writing—review and editing.
A Yonath: conceptualization, supervision, and writing—review and editing.

### Conflict of Interest Statement

The authors declare that they have no conflict of interest.

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
