## [Reviewer comments · Life Science Alliance]

Life Science Alliance

Structural studies on ribosomes of differentially macrolide-resistant *Staphylococcus aureus* strains

Andre Rivalta, Aliza Fedorenko, Alexandre Le Scornet, Sophie Thompson, Yehuda Halfon, Elinor Breiner Goldstein, Sude Çavdaroglu, Tal Melenitzky, Disha Hiregange, Ella Zimmerman, Anat Bashan, Mee-Ngan Yap, and Ada Yonath

DOI: <https://doi.org/10.26508/lsa.202503325>

Corresponding author(s): *Ada Yonath, Weizmann Institute of Science*

Review Timeline:

Submission Date:	2025-03-25
Editorial Decision:	2025-04-29
Revision Received:	2025-05-06
Accepted:	2025-05-07

Scientific Editor: Tim Fessenden

Transaction Report:

April 29, 2025

RE: Life Science Alliance Manuscript #LSA-2025-03325-T

Ada Yonath
The Weizmann Inst. of Science
Department of Structural Biology
Weizmann Inst. of Science
IL-Rehovot 76100
Rehovot 76100
Israel

Dear Dr. Yonath,

Thank you for submitting your manuscript entitled "Structural studies on ribosomes of differentially macrolide-resistant *Staphylococcus aureus* strains". This manuscript was evaluated by two expert reviewers, whose reports are appended to this email.

As you will see, reviewers both concur these findings are important and recommend publication. Reviewer 2 made a few helpful suggestions which should be resolved in a revised manuscript, including a figure depicting SOL and ERY. We would be happy to publish your paper in Life Science Alliance pending these additions and final revisions necessary to meet our formatting guidelines.

- Please upload your main and supplementary figures as single files.
- Please add ORCID ID for the corresponding author -- you should have received instructions on how to do so.
- Please add a Summary Blurb/Alternate Abstract in our system.
- Please add Keywords and a Category for your manuscript in our system.
- Please add the X and Bluesky handles of your host institute/organization, as well as your own and/or one of the authors, to our system.
- Please label Figure S4 correctly.
- The titles in both the system and the manuscript file must be consistent with each other.
- Please be sure that the authorship listing and order are correct.
- Please add your main, supplementary figure, and table legends to the main manuscript text after the references section.
- It is recommended to exclude figures from the manuscript text and upload them separately.
- Please incorporate any points from the Conclusion section into the Discussion; we only allow a Discussion section.
- Please add Author Contributions to our system as well.
- Please add a call-out for Figure 2A-E to your main manuscript text.

LSA now encourages authors to provide a 30-60 second video where the study is briefly explained. We will use these videos on social media to promote the published paper and the presenting author (for examples, see <https://docs.google.com/document/d/1-UWCfbE4pGcDdcgzcmiuJl2XMBJnxKYeqRvLLrLSo8s/edit?usp=sharing>). Corresponding or first-authors are welcome to submit the video. Please submit only one video per manuscript. The video can be emailed to contact@life-science-alliance.org

A. FINAL FILES:

- An editable version of the final text (.DOC or .DOCX) is needed for copyediting (no PDFs).
- High-resolution figure, supplementary figure and video files uploaded as individual files: See our detailed guidelines for

preparing your production-ready images, <https://www.life-science-alliance.org/authors>

B. MANUSCRIPT ORGANIZATION AND FORMATTING:

Sincerely,

Reviewer #1 (Comments to the Authors (Required)):

This study reports cryo-EM structures of the ribosome from a pathogen (*Staphylococcus aureus*), with the antibiotic solithromycin bound. By studying the ribosome from different strains, including those that include methylation as a resistance mechanism, the study finds that SOL can bind to all systems. Further, the structures show minimal changes in the position of SOL, suggesting how this antibiotic can overcome this resistance mechanism. This provides a foundation for further refining this class of molecules, in order to develop more effective antibiotics that overcome known resistance mechanisms. Finally, as a technical point, the study finds that SOL binds differently to the pathogen than in *Thermus* *Thermophilus* ribosomes, which highlights the need to directly study pathogenic systems, as some complexities may be missed in model organisms. The manuscript is well organized and clear. No revisions are necessary.

Reviewer #2 (Comments to the Authors (Required)):

This is an excellent paper that provides new information on macrolide-ribosome interactions by adding 2 new structures and new IC50 values to our accumulating knowledge of macrolide antibiotics. This paper is concise. The authors demonstrate small structural changes to the SOL binding pocket due to double methylation of A2058 which results with higher IC50. SOL possesses an additional alkyl-aryl side chain, which helps the antibiotics bind to the ribosome, even when A2058 is double methylated. This macrolide modification can form a base for the design of future antibiotics that will combat resistant strains of SA.

My comments are:

In introduction, "0.1 million attributable to just MRSA". change to "100,000 attributable to..."

Move the following sentence a bit earlier in the introduction: "Approximately 20-30% of human adults are persistently colonized by *S. aureus*, and carriage in infancy can approach 70% (Piewngam and Otto 2024; Turner et al. 2019). "

In the following sentence, "Multibody focused refinement on the large subunit revealed at 2.08 Å resolution (Supplementary Fig. S5) that the antibiotic remains bound, despite the double methylation of A2058 (Figure 1b).", replace "remains bound" to "can bind in-vitro at high antibiotic concentrations" or "can be accommodated into the ribosome at high concentrations and bind its pocket despite methylation" or similar.

Add a figure that will compare SOL with ERY, 2D and 3D, for the discussion and illustration of their similarities and differences.

Add a reference to the SA structure Eyal et al 2015.

Ada Yonath, Professor

The Martin S. and Helen
Kimmel Professor of
Structural Biology
Director, The Helen and
Milton A. Kimmelman
Center for Biomolecular
Structure and Assembly

Tel: +972 8 934 3028
Ada.yonath@weizmann.ac.il
Department of Chemical and
Structural Biology
The Weizmann Institute of Science
Rehovot 7610001
Israel

05 May, 2025

Dear Editor,

We thank you and the reviewers for evaluation of our work and the constructive feedback, which have improved our manuscript.

We have addressed every point of reviewer 2 in full, as detailed below in red and in the updated manuscript file with track-changes enabled. We also addressed the additional editorial comments, by removing the Conclusions section and incorporating its main points into the Discussion.

During the revision, we noticed that the hydrogen bond between SOL's desosamine O2' and A2058 N6 (Fig. 1, panel B) is not lost but elongated (as can be seen from Fig. 2 panel C). The figure, legend and text have been updated accordingly. This adjustment does not affect any conclusions.

We hope these revisions fully satisfy the reviewer's and editorial requests.

Thank you for your consideration,

Best regards,

Ada Yonath

May 7, 2025

RE: Life Science Alliance Manuscript #LSA-2025-03325-TR

Ada Yonath
Weizmann Institute of Science
Department of Structural Biology
Weizmann Inst. of Science
IL-Rehovot 76100
Rehovot 76100
Israel

Dear Dr. Yonath,

Thank you for submitting your Research Article entitled "Structural studies on ribosomes of differentially macrolide-resistant *Staphylococcus aureus* strains". It is a pleasure to let you know that your manuscript is now accepted for publication in Life Science Alliance. Congratulations on this interesting work.

DISTRIBUTION OF MATERIALS:

Again, congratulations on a very nice paper. I hope you found the review process to be constructive and are pleased with how the manuscript was handled editorially. We look forward to future exciting submissions from your lab.

Sincerely,
